

# An upward continuation method based on spherical harmonic analysis and its application in the calibration of satellite gravity gradiometry data

Qingliang Qu[1,2], Shengwen Yu[1], Guangbin Zhu[2], Xiaotao Chang[2], Miao Zhou[1,2], and Wei Liu[2]
[1] College of Geomatics, Shandong University of Science and Technology, Qingdao 266510, China.
[2] Land Satellite Remote Sensing Application Center of the Ministry of Natural Resources, Beijing 100048,
China.
*Correspondence to:* Shengwen Yu (sdkdswyu@163.com)
**Abstract.** The ground gravity anomalies can be used to calibrate and validate the satellite gravity gradiometry
data. In this study, an upward continuation method of ground gravity data based on spherical harmonic analysis
is proposed, which can be applied to the calibration of satellite observations from the European Space Agency's
Gravity Field and Steady-State Ocean Circulation Explorer (GOCE). Here, the following process was conducted
to apply this method. The accuracy of the upward continuation method based on spherical harmonic analysis was
verified using simulated ground gravity anomalies. The DTU13 global gravity anomaly data were used to
determine the calibration parameters of the GOCE gravitational gradients based on the spherical harmonic
analysis method. The trace and the tensor invariants $I_2$, $I_3$ of the gravitational gradients were used to verify the
calibration results. The results revealed that the upward continuation errors based on spherical harmonic analysis
were much smaller than the noise level in the measurement bandwidth of the GOCE gravity gradiometer. The
scale factors of the $V_{xx}$, $V_{yy}$, $V_{zz}$, and $V_{yz}$ components were determined at an order of magnitude of approximately
$10^{-2}$, the $V_{xz}$ component was approximately $10^{-3}$, and the $V_{xy}$ component was approximately $10^{-1}$. The traces of
gravitational gradients after calibration were improved when compared with the traces before calibration and were
slightly better than the EGG_TRF_2 data released by the European Space Agency (ESA). In addition, the relative
errors of the tensor invariants $I_2$, $I_3$ of the gravitational gradients after calibration were significantly better than
those before calibration. In conclusion, the upward continuation method based on spherical harmonic analysis
could meet the external calibration accuracy requirements of the gradiometer.
**1 Introduction**
The European Space Agency's Gravity Field and Steady-State Ocean Circulation Explorer (GOCE) satellite was
launched on 17 March 2009. The goals of the mission were the retrieval of the global geoid model with 1–2 cm
accuracy and the determination of the global gravity anomalies with 1 mGal accuracy for a spatial resolution of
100 km or less (Drinkwater et al., 2006;Bouman and Fuchs, 2012;van der Meijde et al., 2015;Bouman et al.,
2016;Siemes, 2018). To achieve these goals, the GOCE satellite combined the satellite gravity gradiometry (SGG)
technique with satellite-to-satellite tracking in the high-low mode (SST–hl). The SST technique is sensitive to the
long wavelength signals of the Earth's gravitational field and the SGG technique can contribute to obtaining the
medium and short wavelength signals of the Earth's gravitational field. The electrostatic gravity gradiometer
mounted on the GOCE satellite can measure the second derivative of the Earth's gravitational potential with high
precision. This gradiometer, however, is bandwidth limited to 0.005–0.1 Hz. Therefore, the gravitational gradient
observations may still suffer from system errors, such as scale factors and biases. In this case, an external



calibration strategy is needed to achieve high-precision gravity gradiometry data. In general, the existing Earth
gravitational models, ground gravity data, and SST observations are used to perform the external calibration of
the GOCE gravitational gradients. The calibration of GOCE gravitational gradients using ground gravity data will
be examined and outlined here.
Arabelos and Tscherning (Arabelos and Tscherning, 1998) described a simulation study of the external calibration
approach for SGG data with ground gravity data and used the least-squares collocation (LSC) method to detect
the systematic errors of gravitational gradients. The a priori covariance relationship of the upward continuation
of ground gravity data onto gravitational gradients was discussed in Bouman et al. and Pail (Pail, 2002;Bouman
and Koop, 2003). Denker (Denker, 2002) applied the least squares spectral combination technique to the upward
continuation of the ground gravity data onto gravitational gradients at satellite altitude. It was proven that the
accuracy of this method can reach a few mE (1 mE = $10^{-12}/s^2$). Two methods for the upward continuation of
ground gravity data onto gravitational gradients, namely, the LSC and integral formula methods based on the
spectral combination technique, were discussed and compared in Wolf and Denker (Wolf and Denker, 2005). A
synthetic geopotential model, which combined the GRACE geopotential model, EGM96 geopotential model, and
GPM98C geopotential model, was used to simulate the gravity anomalies on terrain and on an ellipsoid. This
study revealed that the results of the two methods were similar and the accuracies of six components were 0.1–
0.6 mE and 0.3–1.4 mE, respectively, when the gravity anomalies on the terrain and ellipsoid were applied for the
continuation. The integral formulas based on the extended Stokes and Hotine formulas were used by Kern and
Haagmans (Kern and Haagmans, 2005) to determine all the components of the gravitational gradients from
terrestrial gravity data. They found that the difference between the computed gravitational gradients and the model
values from the GPM98A geopotential model ranged from 1.5 to 2.5 mE for all components. To validate the SGG
data, an external calibration model based on the regional ground gravity data was described in Bouman et al.
(Bouman et al., 2004;Bouman et al., 2009;Bouman et al., 2011). The results showed that the scale factors of the
gradiometer can be determined at the $10^{-2}$ level using the LSC upward continuation method, and that ground
gravity data can be used to validate the measured and calibrated gravitational gradients. A least squares
modification of the extended Stokes formula and its second-order radial derivative was proposed by Eshagh
(Eshagh, 2010). This method was used to generate the gravitational gradients at satellite altitude from the ground
gravity data to validate the SGG data. The airborne gravity data of the Antarctic region were applied to validate
the GOCE gravity gradiometry data in Yildiz et al. (Yildiz, 2012;Yildiz et al., 2016). They concluded that the
differences between the calculated gravitational gradients from the LSC upward continuation method and the
GOCE gravitational gradient observations were 9.9 mE, 11.5 mE, 11.6 mE, and 10.4 mE in the high-precision
components $V_{xx}$, $V_{yy}$, $V_{zz}$, and $V_{xz}$, respectively. The validation of the $V_{zz}$ component of the GOCE gravitational
gradients by geoidal undulation using semi-stochastic modifications of the Abel-Poisson integral was discussed
in Eshagh (Eshagh, 2011). Šprlák et al. (Šprlák et al., 2015) presented new integral transforms of the gravitational
potential disturbances derived from satellite altimetry data onto the gravitational gradients at satellite altitude.
Thus, we see that the LSC and integral formula methods are commonly used in the upward continuation of the
ground gravity data onto the gravitational gradients at satellite altitude for the calibration of SGG data. The key
to applying the LSC method is to construct the covariance functions between the gravity anomalies and the
gravitational gradients. The inverse matrix of the large covariance matrix is very difficult to solve in massive data





processing, however. The calculation of the transformation kernel function in the integral formula method is
relatively complicated, and the influence of the boundary effect should be considered.

In this article, we discuss the possibilities of spherical harmonic analysis for the upward continuation of the ground
gravity data onto the gravitational gradients at satellite altitude. The upward continuation method based on
spherical harmonic analysis is more convenient to use than the LSC and integral formula methods. In addition,
the DTU13 gravity anomalies were used to calibrate the GOCE SGG data based on this method.
**2 Methods**
**2.1 Upward continuation method based on the spherical harmonic analysis**
A square integrable function $f(\theta, \lambda)$ defined on the unit sphere can be expanded into a series of spherical
harmonics as (Colombo, 1981; Kern, 2003):

$$f(\theta, \lambda) = \sum_{n=0}^{\infty} \sum_{m=0}^{n} \left[ \bar{C}_{nm} \cos m\lambda + \bar{S}_{nm} \sin m\lambda \right] \bar{P}_{nm}(\cos \theta) , \tag{1}$$

where $\theta, \lambda$ are the geocentric co-latitude and longitude of the computation point, respectively, $\bar{P}_{nm}$ is the fully
normalized Legendre polynomial of degree $n$ and order $m$, and $\bar{C}_{nm}$ and $\bar{S}_{nm}$ denote the fully normalized
gravity field harmonic and Stokes coefficients, respectively.
The purpose of spherical harmonic analysis is to estimate the coefficients $\bar{C}_{nm}$ and $\bar{S}_{nm}$ based on the function
$f(\theta, \lambda)$, which is the inverse process of spherical harmonic synthesis. Therefore, the coefficients can be obtained
using:

$$\left. \begin{matrix} \bar{C}_{nm} \\ \bar{S}_{nm} \end{matrix} \right\} = \frac{1}{4\pi} \int_{\sigma} f(\theta, \lambda) \left\{ \begin{matrix} \cos m\lambda \\ \sin m\lambda \end{matrix} \right\} \bar{P}_{nm}(\cos \theta) d\sigma , \tag{2}$$

where $d\sigma$ is the grid area and $d\sigma = \sin \theta d\theta d\lambda$. In general, the function $f(\theta, \lambda)$ is unknown, but we can obtain
the values of each grid point or the average values over the grid areas. Thus, Equation (2) can be discretized as:

$$\left. \begin{matrix} \bar{C}_{nm} \\ \bar{S}_{nm} \end{matrix} \right\} = \frac{1}{4\pi} \sum_{i=1}^{N} \sum_{j=1}^{2N} f(\theta_i, \lambda_j) \left\{ \begin{matrix} \cos m\lambda_j \\ \sin m\lambda_j \end{matrix} \right\} \bar{P}_{nm}(\cos \theta_i) \sin \theta_i \Delta_{ij} , \tag{3}$$

where $\Delta_{ij} = \Delta\theta \cdot \Delta\lambda$ and $\Delta\theta = \Delta\lambda$ when the grid is regular, and $N$ is the number of latitude grid points.
In this study, the spherical harmonic analysis of gravity anomalies was needed. The gravity anomaly can be
computed as:

$$\Delta g = \frac{GM}{r^2} \sum_{n=2}^{n_{max}} \sum_{m=0}^{n} (\frac{R}{r})^n (n-1) \left( \bar{C}_{nm}^* \cos m\lambda + \bar{S}_{nm} \sin m\lambda \right) \bar{P}_{nm}(\cos \theta) , \tag{4}$$

where $\Delta g$ is the gravity anomaly, $M$ is the mass of the Earth, $G$ is the gravitational constant, $R$ is the mean
equatorial radius, $r$ is the geocentric radius, and $\bar{C}_{nm}^*$ is the spherical harmonic coefficients from which the
normal ellipsoid gravitational potential coefficients have been subtracted.
Combining Equations (3) and (4), the point values of the spherical harmonic analysis expression of the gravity
anomaly can be derived as:



$$\left.\begin{array}{l}\overline{C}_{nm}^{*} \\ \overline{S}_{nm}\end{array}\right\} = \frac{r^2}{4\pi GM(n-1)}(\frac{r}{R})^n \sum_{i=1}^{N}\sum_{j=1}^{2N}\Delta g(\theta_i,\lambda_j)\sin\theta_i\Delta_{ij}.$$ (5)

The fully normalized gravity field harmonic coefficients $\overline{C}_{nm}$ can be obtained by adding the normal ellipsoid
gravitational potential coefficients to $\overline{C}_{nm}^{*}$. Combining with the precise scientific orbit data of the GOCE, the
ground gravity anomalies can be upwardly continued onto the gravitational gradients at satellite altitude in the
local north-oriented frame (LNOF), where the x-axis points to the north, the y-axis east, and the z-axis radially
outward (Figure 1).

### 2.2 External calibration method


The calibration using ground gravity data relies on the comparison of the GOCE gravitational gradients and the
upward continuation gravitational gradients. The GOCE gravitational gradients from the EGG_NOM_2 data are
presented in the gradiometer reference frame (GRF), where the x-axis is parallel to the instantaneous direction of
the orbital velocity vector, and the y-axis is parallel to the instantaneous direction of the orbital angular momentum
(Figure 1). The upward continuation gradients are generally expressed in the LNOF, however. Therefore, frame
transformation is required during the external calibration process. When the gravitational gradients in the LNOF
are converted to the GRF, several coordinate rotation steps are necessary. The model of the frame transformation
is:
$$V_{GRF} = RV_{LNOF}R^{T},$$ (6)

where $R$ is the transformation matrix, such that $R = R_{EFRF}^{LNOF}\cdot R_{IRF}^{EFRF}\cdot R_{GRF}^{IRF}$ (Fuchs and Bouman, 2011). $R_{EFRF}^{LNOF}$ is
the transformation matrix from the LNOF to the Earth-fixed reference frame (EFRF) system, where the x-axis is
fixed in the equatorial plane in the direction of the Greenwich meridian, and the z-axis is the direction of the pole
(Figure 1); $R_{IRF}^{EFRF}$ is the transformation matrix from the EFRF to the inertial reference frame (IRF), where the x-
axis is fixed in the equatorial plane in the direction of the vernal equinox, and the z-axis is the direction of the
pole (Figure 1); and $R_{GRF}^{IRF}$ is the transformation matrix from the IRF to the GRF.
The calibration parameters of the GOCE gravitational gradients were determined as follows:
$$V_{ij}^{m}(t) = \lambda V_{ij}^{s}(t) + b \quad i,j = x,y,z,$$ (7)

where $V_{ij}^{m}$ are the upward continuation values, $V_{ij}^{s}$ are the GOCE gradiometry observations, $\lambda$ is the scale factor,
$b$ is the bias, and $t$ is the time. Least squares estimation was then used to estimate the parameters.





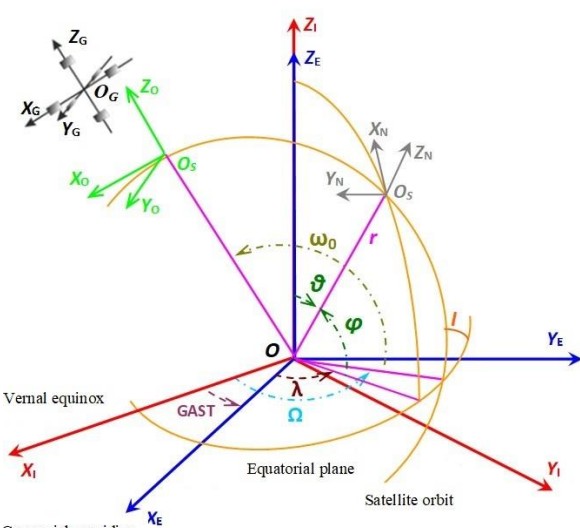


Figure 1. Reference systems for the GOCE satellite: $O_G - X_G Y_G Z_G$ is the GRF coordinate system, $O_S - X_N Y_N Z_N$ is the
LNOF coordinate system, $O - X_I Y_I Z_I$ is the IRF coordinate system, and $O - X_E Y_E Z_E$ is the EFRF coordinate system.

## 3 Results and Discussion

### 3.1 Accuracy of the upward continuation method

The accuracy of the upward continuation method based on spherical harmonic analysis was verified by simulation.
The high-precision global gravity field model EGM2008 (Pavlis et al., 2008) was selected for the computation.
The precise orbital data of the GOCE with a time interval of 1 s from 11 February 2011 to 17 February 2011 were
used to compute the gravitational gradients at satellite altitude. Hence, the total number of observations was
604,800. The verification schemes were designed as follows.
Scheme 1. The grid gravity anomalies $\Delta g^{true}$ with a resolution of 0.5° on the sphere calculated by the EGM2008
field to degree and order 360 were regarded as the simulated ground gravity data. Next, combining with the precise
orbit data, the simulated ground gravity data $\Delta g^{true}$ were upwardly continued onto the gravitational gradients
$V_{ij}^0 (i, j = x, y, z)$ at the satellite altitude based on spherical harmonic analysis. Finally, the upward continuation
gravitational gradients $V_{ij}^{ture}$ were compared with the gravitational gradients   directly calculated by the EGM2008
field. The upward continuation errors based on spherical harmonic analysis could be obtained using this scheme.
Scheme 2. The gravity anomalies $\Delta g^{true}$ were added to 5 μGal, 1 mGal and 2 mGal white noise, serving as the
ground-truth measurement data. The remaining steps were the same as scheme 1. The influence of the accuracy
of ground gravity anomalies on upward continuation errors could be obtained by this scheme.
Figure 2 shows the spatial distribution of upward continuation errors using different ground gravity accuracy in
scheme 2. There is no general pattern can be observed, indicating that the upward continuation errors were



randomly distributed over the orbits. When the accuracy of the ground gravity anomalies was 5 μGal, the
differences between the upward continuation gravitational gradients and the gravitational gradients calculated by
the EGM2008 field for all of the components ranged from -0.4 to 0.4 mE. When the accuracy of the ground gravity
anomalies was 1 mGal and 2mGal, the differences mostly varied from -4 to 4 mE and -6 to 6 mE. Therefore, the
accuracy of the ground gravity anomalies exerted a significant influence on the upward continuation errors. Table
1 lists the statistics of the upward continuation errors in each component of the gravitational gradients for the
different schemes. The accuracy of the upward continuation of the $V_{zz}$ component was lower than that of the other
components. When there was no noise in the gravity anomalies (scheme 1), the errors caused by the upward
continuation method based on spherical harmonic analysis were $10^{-3}$ mE in the $V_{xx}$, $V_{yy}$, $V_{zz}$, and $V_{xz}$ components
and $10^{-5}$ mE in the $V_{xy}$ and $V_{yz}$ components. Meanwhile, the noise level was approximately 5–8 mE in the
measurement bandwidth of the gravity gradiometer (Rummel et al., 2011). Thus, it can be seen that the upward
continuation errors were far less than the noise level in the measurement bandwidth of the gradiometer. When the
gravity anomalies contained 5 μGal of white noise, the standard deviations of the upward continuation errors of
the $V_{xx}$, $V_{yy}$, $V_{xy}$, $V_{xz}$, and $V_{yz}$ components were $10^{-2}$ mE and 0.1 mE in the $V_{zz}$ component, which were still
significantly lower than the noise level in the measurement bandwidth of the gravity gradiometer. When the
gravity anomalies contained 1 mGal or 2 mGal of white noise, the standard deviations of the upward continuation
errors of all components ranged from 0.5 to 1.6 mE, which was also less than the noise level in the measurement
bandwidth of the gravity gradiometer. This indicates that the upward continuation method for gravity anomalies
of gravitational gradients based on spherical harmonic analysis can be used to calibrate the SGG data. Moreover,
if the ground data are more accurate, then the gravitational gradients at the satellite altitude obtained by upward
continuation will also be more accurate.





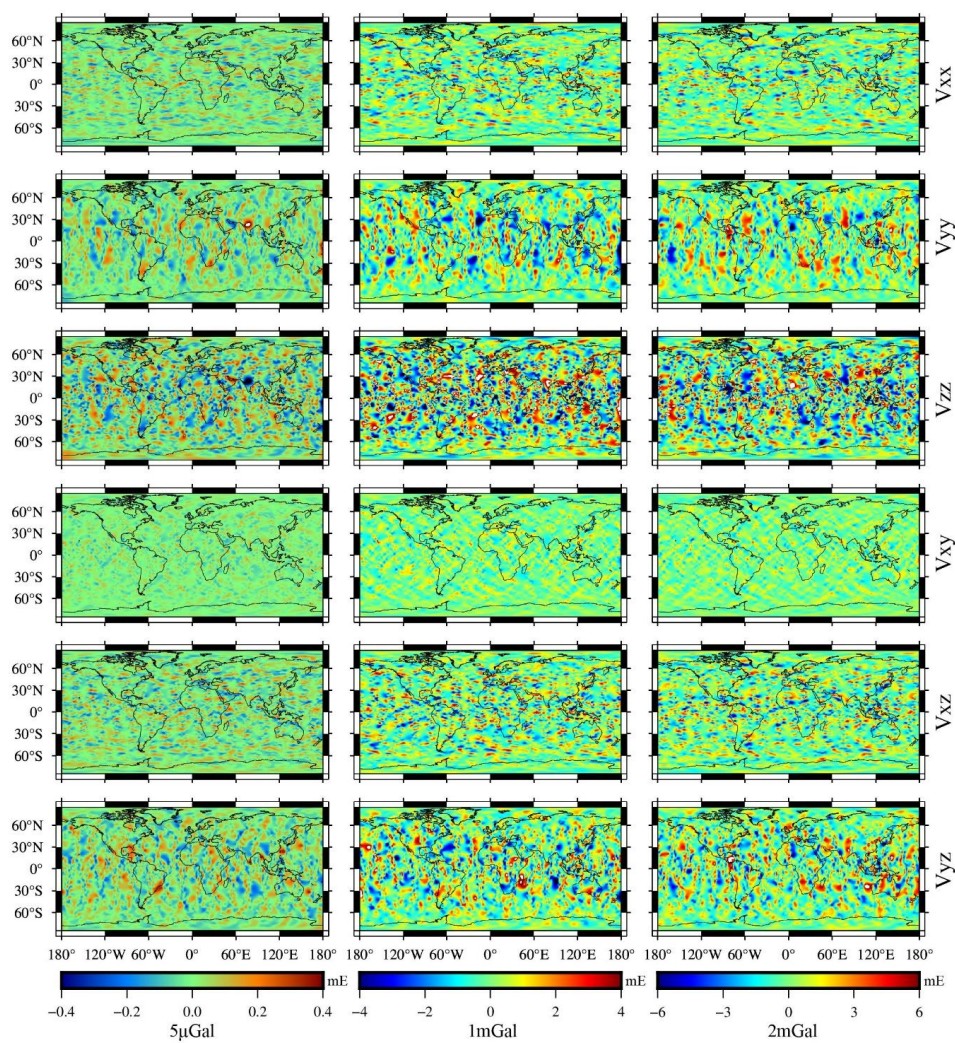

Figure 2. Distribution of upward continuation errors of the gravitational gradients using ground gravity data.

Table 1. Standard deviation of upward continuation errors in gravitational gradients for different simulation schemes (mE).

| Component | | $V_{xx}$ | $V_{yy}$ | $V_{zz}$ | $V_{xy}$ | $V_{xz}$ | $V_{yz}$ |
|---|---|---|---|---|---|---|---|
| Scheme 1 | | $1.4\times10^{-3}$ | $1.0\times10^{-3}$ | $2.5\times10^{-3}$ | $2.8\times10^{-5}$ | $1.7\times10^{-3}$ | $4.6\times10^{-5}$ |
| Scheme 2 | 5 μGal | $7.2\times10^{-2}$ | $7.2\times10^{-2}$ | $1.0\times10^{-1}$ | $4.0\times10^{-2}$ | $8.3\times10^{-2}$ | $8.3\times10^{-2}$ |
| | 1 mGal | 0.8 | 0.8 | 1.2 | 0.5 | 1.0 | 1.0 |
| | 2 mGal | 1.0 | 1.0 | 1.6 | 0.7 | 1.3 | 1.3 |





### 3.2 Calibration results with DTU13 global gravity anomalies

Compared with the global gravity field model, the DTU13 (Andersen et al., 2014;Andersen et al., 2015) gravity
anomalies contain more high frequency signals, and its accuracy is about 2 mGal. Therefore, the DTU13 global
gravity anomalies with a resolution of 0.5° were applied in the numerical experiment to calibrate the gravitational
gradients of the GOCE satellite. The DTU13 global gravity anomalies are shown in Figure 3. To reduce the
influence of the long wavelength signals of the gravitational field, the remove-restore procedure was applied based
on the reference geopotential model EGM2008 up to degree and order 360. The upward continuation of the
residual gravity was extended to the satellite altitude using the spherical harmonic analysis method, and the long
wavelength signals of the gravity field were then restored. The GOCE data used in this study spanned the period
February 11 to June 23, 2011, with a time interval of 1 s. Referring to Bouman et al. (Bouman et al., 2011), the
calibration period was set to 7 days. Hence, the data were divided into 19 weeks.

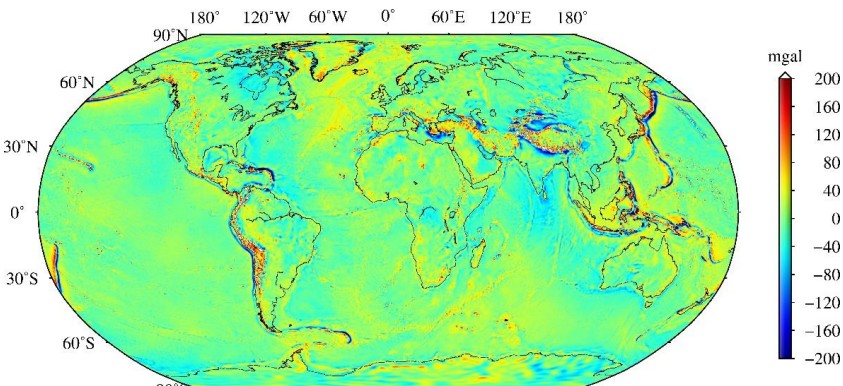

Figure 3. Global gravity anomalies of the DTU13.

Corrections for temporal gravity field variations and outlier detection of the GOCE gravitational gradients were
conducted in the EGG_NOM_2 file. The outliers were replaced by cubic spline interpolation values in this study.
The power spectral density (PSD) of the GOCE gravitational gradients and the upward continuation values from
the ground gravity anomalies are displayed in Figures 4(a) and (b). Because of the measurement bandwidth
limitation of the gravity gradiometer, the noise of the gravitational gradients was large below the lower limit of
the measurement bandwidth, which exhibited a $1/f$ behavior. Therefore, a second-order high-pass Butterworth
filter was adopted before calibration. Various filter cut-off frequencies were discussed in Bouman et al. (Bouman
et al., 2011). They pointed out that the cut-off frequencies of 3, 5, and 7 mHz are appropriate for GOCE SGG
data. Therefore, 3 mHz was used as the cut-off frequency in this study, which was below the lower bound of the
measurement bandwidth and retained more gravitational gradient signals of the GOCE SGG data. Figures 4(c)
and (d) are the filtered signals of the GOCE gravitational gradients and the upward continuation values. It is clear
that the effect of low frequency signals was suppressed, although the noise level was still high when the frequency
was close to the lower limit of the measurement bandwidth. When the frequency was between 0.005 Hz and 0.03
Hz, the GOCE gravitational gradients in the $V_{xx}$, $V_{yy}$, $V_{zz}$, and $V_{xz}$ components decreased rapidly, while the $V_{xy}$ and
$V_{yz}$ components remain a constant about $10^3$ mE. Meanwhile, the upward continuation values decreased rapidly





in all six components. When the frequency was between 0.03 Hz and 0.1 Hz, the $V_{xx}$, $V_{yy}$, $V_{zz}$, and $V_{xz}$ components
decreased to 10–20 mE for the GOCE gravitational gradients, although the $V_{xx}$, $V_{yy}$, $V_{zz}$, and $V_{xy}$ components
decreased to approximately $10^{-2}$ mE and the $V_{xz}$, $V_{yz}$ components decreased to approximately 1 mE for the upward
continuation values.

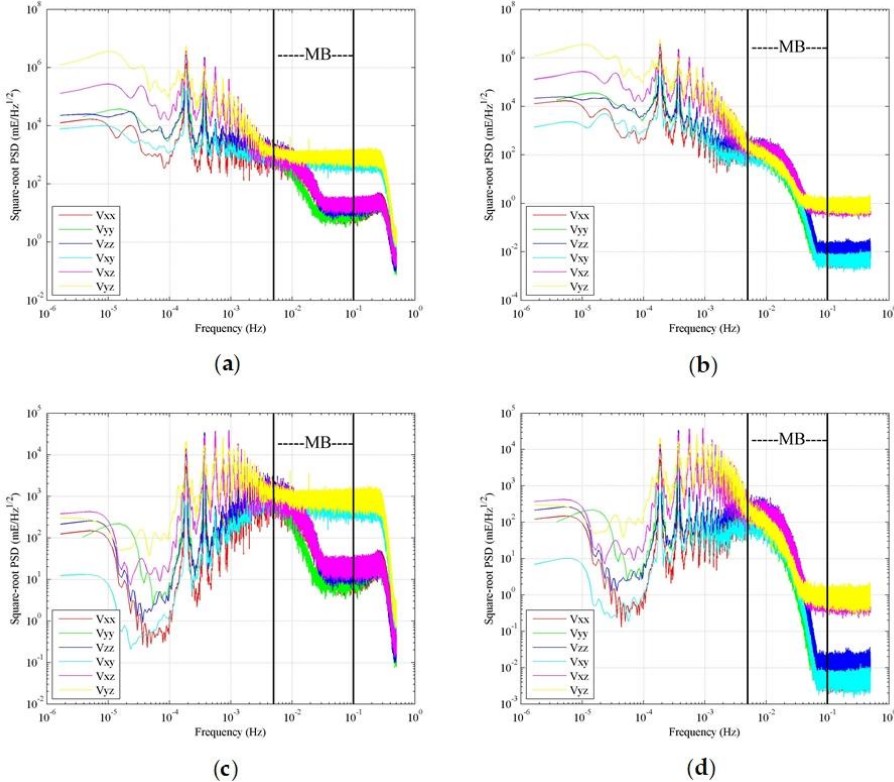

Figure 4. Power spectral density of gravitational gradients: (a) GOCE observations; (b) upward continuation
values; (c) GOCE observations after high-pass filtering; (d) upward continuation values after high-pass filtering.
Figures 5 and 6 reflect the changes of the scale factors and biases for the GOCE gravitational gradients. It appears
that the scale factors had a period of approximately 3 weeks, which corresponds to the 20-day subcycle of the
GOCE satellite orbit. After high-pass filtering, the biases were very small, with maxima on the order of $10^{-5}$ for
all components of the GOCE gravitational gradients. Table 2 lists the statistics of the scale factors for the six
components of the gravitational gradients. The deviations between the mean values of the scale factors and one
ranged from approximately 0.02 to 0.03 for the diagonal components. These results are larger than those of
Veicherts et al. (Veicherts et al., 2011) for Australia, Canada, and parts of Scandinavia, but smaller than those of
the Norway area. The reason for these differences is that, on the one hand, the accuracy levels of the DTU13
gravity anomalies and the regional ground gravity data used in the Veicherts study are different. On the other
hand, the calibration parameters are determined globally rather than in a certain area. The stability of the scale





factors for the diagonal components had a magnitude of approximately $10^{-2}$, while the ultra-sensitive component
$V_{xz}$ was the best, reaching a magnitude of $10^{-3}$. In contrast, the stability of the scale factor for the $V_{xy}$ component
was poor, only about $10^{-1}$. Given the scale factors derived from the comparison between the filtered upward
continuation gravitational gradients and the filtered GOCE gravitational gradients, the upward continuation
gravitational gradients were regarded as the true values. The $V_{xy}$ component exhibited the maximum difference
between the GOCE gravitational gradients and the upward continuation values within the measurement bandwidth,
as seen in Figures 4(c) and (d). In other words, the noise level was highest in the $V_{xy}$ component, so the scale
factors of this component were unstable. This phenomenon is also consistent with the design characteristics of the
GOCE gravity gradiometer.

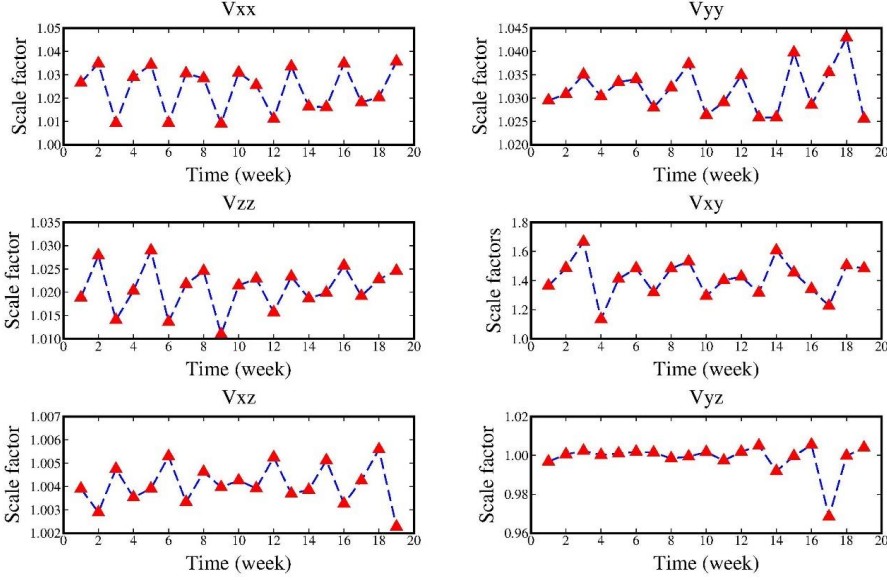


Figure 5. Variations of scale factors during the calibration period.





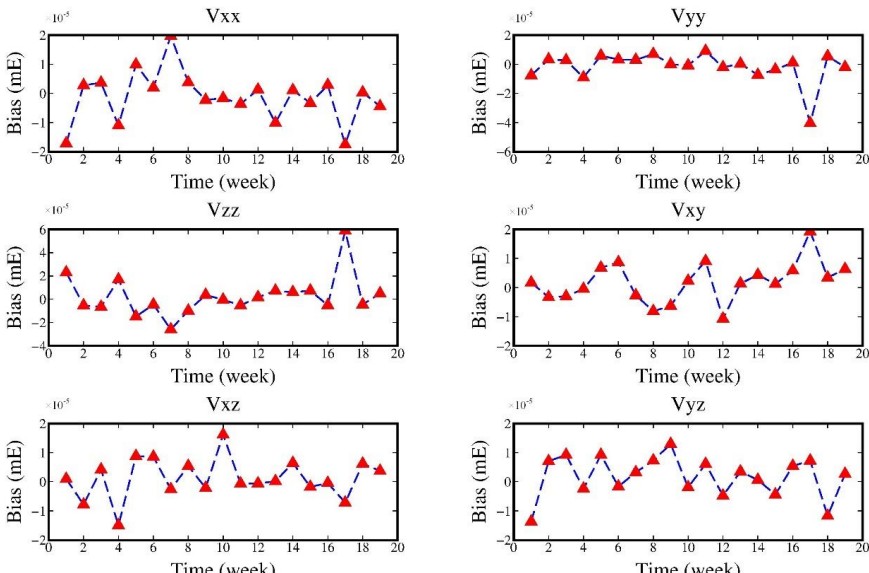


Figure 6. Variations of biases during the calibration period.

Table 2. Statistics of the scale factors.

| Component | Minimum | Maximum | Mean | Standard deviation |
|-----------|---------|---------|------|--------------------|
| $V_{xx}$ | 1.0090 | 1.0357 | 1.0239 | $9 \times 10^{-3}$ |
| $V_{yy}$ | 1.0256 | 1.0430 | 1.0318 | $5 \times 10^{-3}$ |
| $V_{zz}$ | 1.0110 | 1.0290 | 1.0208 | $5 \times 10^{-3}$ |
| $V_{xy}$ | 1.1348 | 1.6660 | 1.4177 | $1 \times 10^{-1}$ |
| $V_{xz}$ | 1.0023 | 1.0056 | 1.0041 | $8 \times 10^{-4}$ |
| $V_{yz}$ | 0.9684 | 1.0055 | 0.9988 | $8 \times 10^{-3}$ |

**4 Discussion of the calibration results**

After the calibration of the GOCE gravitational gradients was completed, the calibration results needed to be

verified and analyzed to ensure the calibration accuracy, which was key to checking the quality of the gravitational

gradients of the GOCE satellite.

(1)  Verification by the trace-free characteristics of gravitational gradients

The gravitational gradients satisfy the Laplace equation in the space around the Earth, i.e., the trace of the

gravitational gradient observations is 0. Based on this criterion, the calibration results were verified and evaluated.

The calibration parameters were applied to the high-pass-filtered SGG observations, after which the trace of the

gravitational gradients following calibration could be obtained. Figure 7 displays the root mean square error of

the trace of the gravitational gradients in the 19 weeks after calibration. It is obvious that the trace of the GOCE

gravitational gradients improved after calibration. During the calibration period, the maximum value appeared in

the ninth week, at which point it was approximately 23.2 mE before calibration and approximately 22.7 mE after





calibration. Although the GOCE satellite was not offline and its onboard system was operating normally at that
time, a large number of GOCE gravitational gradient data were still missing. This phenomenon may be related to
changes in the external space environment of the electrostatic gravity gradiometer, such as solar and geomagnetic
activities.

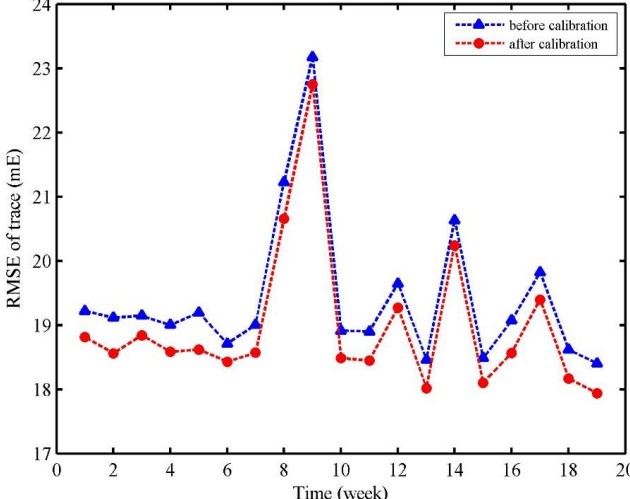


Figure 7. Root mean square error of the trace of the gravitational gradients before and after calibration.
In addition, the calibrated gravitational gradients in the EGG_TRF_2 (ESA, 2014) were used to verify the results.
The EGG_TRF_2 observations were transferred from the LNOF to the GRF and filtered by the same high-pass
filter described in Section 3.2. The time-dependent change of the trace between the calibrated GOCE gravitational
gradients and the EGG_TRF_2 observations, along with the histogram of residuals in 1 day, are shown in Figure
8. From a time series perspective, the trace of the calibrated GOCE gravitational gradients was consistent with
the EGG_TRF_2 data. The histogram shows that 95% of the differences between the calibrated GOCE
gravitational gradients and the EGG_TRF_2 observations were within 5 mE and the standard deviation of the
residuals was approximately 2.3 mE. The standard deviation of the trace of the calibrated gradiometry
observations in this study was approximately 18.6 mE, whereas the EGG_TRF_2 was approximately 18.9 mE.
This indicates that the accuracy of the calibration results of the gravitational gradients based on spherical harmonic
analysis was slightly better than that of the EGG_TRF_2 data.





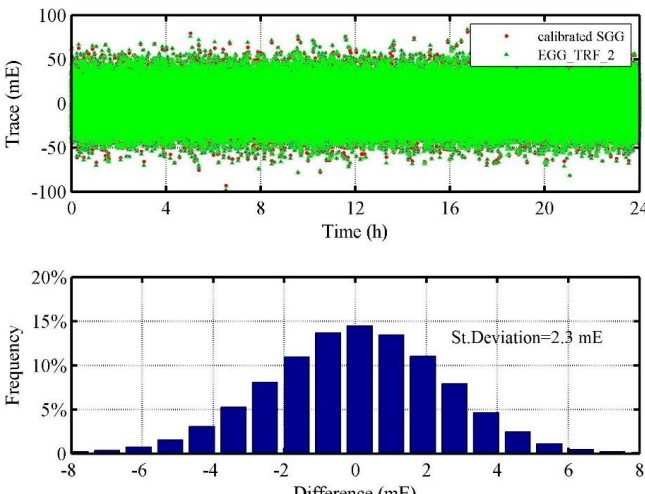


Figure 8. Comparison of the trace of the calibrated gravitational gradients with the trace of the EGG_TRF_2.
(2)  Verification by the tensor invariants method
Because the trace criterion can only verify the overall accuracy of the calibrated diagonal components of the
gravitational gradients, the tensor invariants were introduced into the accuracy verification process. Combined
with the prior gravity field model information, the independent accuracy verification of the diagonal component
and the non-diagonal component of the gravitational gradients could be realized.
The application of 3 tensor invariants in the verification of the gravitational gradients can be expressed as (Baur
et al., 2008;Lu et al., 2018):
$$\begin{cases} I_1 = V_{xx} + V_{yy} + V_{zz} \\ I_2 = -\dfrac{1}{2} V_{xx}^2 + V_{yy}^2 + V_{zz}^2 - V_{xy}^2 - V_{xz}^2 - V_{yz}^2 \\ I_3 = V_{xx}V_{yy}V_{zz} + 2V_{xy}V_{xz}V_{yz} - V_{xx}V_{yz}^2 - V_{yy}V_{xz}^2 - V_{zz}V_{xy}^2 \end{cases}.$$
(8)

It is clear that the tensor invariant $I_1$ is the trace of the gravitational gradients, which was utilized before. The
tensor invariants $I_2$ and $I_3$ comprise all six components of the gravitational gradients, and their relative error before
and after calibration (Equation [9]) could be used to evaluate the calibration results,
$$\delta_2^o = \frac{\left| I_2^o - I_2^r \right|}{I_2^r} \times 100\%$$

$$\delta_2^c = \frac{\left| I_2^c - I_2^r \right|}{I_2^r} \times 100\%$$

$$\delta_3^o = \frac{\left| I_3^o - I_3^r \right|}{I_3^r} \times 100\%$$

$$\delta_3^c = \frac{\left| I_3^c - I_3^r \right|}{I_3^r} \times 100\%$$
(9)





The superscripts *o*, *c*, and *r* represent the GOCE gravitational gradient observations, the calibrated gravitational
gradient values, and the model values calculated by the EGM2008 gravitational potential model up to degree and
order 360, respectively. Here, the calibrated gravitational gradient values indicate that the signals below 3 mHz
were replaced by the signals from the EGM2008 gravitational potential model up to degree and order 360.
Therefore, $\delta_2^o, \delta_3^o$ are the tensor invariants $I_2, I_3$ before calibration, whereas $\delta_2^c, \delta_3^c$ are the tensor invariants
$I_2, I_3$ after calibration.
The statistics for the relative errors of the tensor invariants $I_2, I_3$ before and after calibration in the first calibration
period are listed in Table 3. For the $V_{xx}$, $V_{yy}$, $V_{zz}$, and $V_{xz}$ components, the relative errors of the tensor invariants
$I_2, I_3$ after calibration were 2–4 orders of magnitude smaller than those before calibration. For the less accurate
components $V_{xy}$ and $V_{yz}$, the effects of calibration were more apparent. This indicates that the calibration result of
the upward continuation method based on spherical harmonic analysis was effective when the tensor invariant $I_2$
or $I_3$ was used to verify the calibration accuracy.
Table 3. Relative errors of tensor invariant $I_2$ (%).

| Component | | $V_{xx}$ | $V_{yy}$ | $V_{zz}$ | $V_{xy}$ | $V_{xz}$ | $V_{yz}$ |
|---|---|---|---|---|---|---|---|
| Invariant $I_2$ | Before calibration | $3.1\times10^{-2}$ | 0.2 | $4.2\times10^{-2}$ | 1.7 | $1.6\times10^{-2}$ | 156.1 |
| | After calibration | $1.6\times10^{-4}$ | $1.5\times10^{-4}$ | $3.7\times10^{-4}$ | $3.8\times10^{-6}$ | $5.7\times10^{-6}$ | $3.3\times10^{-4}$ |
| Invariant $I_3$ | Before calibration | 0.2 | 0.3 | $3.4\times10^{-2}$ | 4.95 | $2.4\times10^{-2}$ | 227.65 |
| | After calibration | $4.9\times10^{-4}$ | $4.5\times10^{-4}$ | $2.8\times10^{-4}$ | $9.8\times10^{-6}$ | $8.5\times10^{-6}$ | $5.0\times10^{-4}$ |

**5 Conclusions**
Based on the spherical harmonic analysis method, the gravitational gradients at the altitude of the GOCE satellite
were calculated using the simulated ground gravity anomaly data, and verification was performed. The external
calibration parameters of the GOCE gravitational gradients were determined using DTU13 global gravity
anomalies.
The simulation process verified the accuracy and application potential for calibrating the satellite gravity
gradiometry data using the spherical harmonic analysis method. The results revealed that the upward continuation
errors were smaller than the noise level in the measurement bandwidth of the gravity gradiometer.
After calibrating the GOCE gravitational gradients with the DTU13 ground gravity data, the stability of the scale
factors in the $V_{xx}$, $V_{yy}$, $V_{zz}$, and $V_{yz}$ components had a magnitude of approximately $10^{-2}$, and approximately $10^{-3}$ in
the $V_{xz}$ component, whereas the stability of the $V_{xy}$ component had a magnitude of only $10^{-1}$. The reliability of the
calibration results was verified through the gravitational gradients trace and the tensor invariants method. The
trace of the gravitational gradients after calibration was smaller than before calibration, with an average value of
18.6 mE after calibration, which was slightly better than the accuracy of the EGG_TRF_2 data. The relative errors
of the tensor invariants $I_2, I_3$ after calibration were 2–4 orders of magnitude smaller than the errors before
calibration.

**Data Availability:** The satellite gravity gradiometry data used in this study are available from https://goce-ds.eo.esa.int/oads/access/collection and the DTU13 gravity anomaly data are available from ftp://ftp.spacecenter.dk/pub/.

**Conflicts of Interest:** The authors declare that there are no conflicts of interest regarding the publication of this paper.

**Author Contributions:** Conceptualization, Qingliang Qu, Guangbin Zhu and Xiaotao Chang; methodology, Qingliang Qu and Guangbin Zhu; software, Qingliang Qu; validation, Qingliang Qu; formal analysis, Qingliang Qu; investigation, Qingliang Qu; resources, Miao Zhou and Wei Liu; data curation, Miao Zhou and Wei Liu; writing–original draft preparation, Qingliang Qu; writing–review and editing, Shengwen Yu, Guangbin Zhu and Xiaotao Chang; visualization, Shengwen Yu, Guangbin Zhu and Xiaotao Chang; supervision, Shengwen Yu, Guangbin Zhu and Xiaotao Chang; projectadministration, Guangbin Zhu; funding acquisition, Guangbin Zhu. All authors have read and agree to the published version of the manuscript.

**Funding Statement:** This research was funded by the Project of Civil Aerospace Advanced Research (Grant No. D010103), Major Project of High Resolution Earth Observation System (Construction and Application Technology of GF-7 Satellite Elevation Datum Conversion Model, Grant No. 42-Y20A09-9001-17/18), Open Funding of the Key Laboratory of Surveying and Mapping Science and Geospatial Information Technology of Ministry of Natural Resources (Grant No. 201907), the Operation and Maintenance Project of Land Satellite Remote Sensing Application System, MNR (Grant No. AB1901), and the Key Laboratory of the Geospace Environment and Geodesy, Ministry of Education, Wuhan University (Grant No. 18-01-05).

**Acknowledgments:** The European Space Agency is acknowledged for kindly providing the GOCE gravitational gradients. The Technical University of Denmark is acknowledged for kindly providing the DTU13 gravity anomalies.

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
