# Peer review of "An upward continuation method based on spherical harmonic analysis and its application in the calibration of satellite gravity"

_Solid Earth, 2020_

## Referee Comment (RC1)

**Review of the manuscript (se-2020-201)**
**"An upward continuation method based on spherical harmonic analysis and its application in the calibration of satellite gravity gradiometry data"**

Dear authors and dear editor,

please find below the review of the manuscript se-2020-201.

**1 General comments**

The paper "*An upward continuation method based on spherical harmonic analysis and its application in the calibration of satellite gravity gradiometry data*" by Qingliang Qu, Shengwen Yu, Guangbin Zhu, Xiaotao Chang, Miao Zhou and Wei Liu studies the effect of an external calibration of gravity gradients as measured by the satellite mission GOCE. For that purpose, ground gravity data products – i.e. global gridded datasets – are upward continued and converted to gravity gradients in the reference frame of the gradiometer on-board the GOCE satellite. For the change of functional and the upward continuation of the ground gravity reference dataset, the authors propose a conversion to a spherical harmonic series. Within data windows of a fixed length, biases and scale factors are estimated to externally calibrate the measurements with respect to the reference dataset. The trace-free condition as well as rotational invariants $I_2$ and $I_3$ are used to access the quality of the calibration. The authors validate the calibration computing the trace reduction as well as the consistency of the higher order invariants with respect to the derived reference gradients.

In the current version of the manuscript, I see major issues in all main parts, i.e. the Introduction and Motivation, the Methods section, the Results and Discussion part as well as the final Discussion of the results and the derived Conclusions. For that reason I have to suggest to *reject* the submitted manuscript.

In the first part (i.e. 1. Introduction) I am missing the motivation for the study. It remains open, why there is a need for new, respectively alternative, calibration procedures compared to the existing and published approaches. Although there have been several major updates in GOCE L1B gravity gradient processing (Siemes et al., 2019), which might result in different conclusions for the external calibration compared to the existing published studies, the authors do not address the recent GOCE L1B updates and do not relate their study to the improved L1B data.

The methods section summerizes the upward continuation via a conversion of global gridded gravity data sets to spherical harmonics, for which well established procedures exist. The benefit of using global gridded data set, compared to directly an existing spherical harmonic model, remains open and is not shown. Furthermore, the presentation of the conversion to spherical harmonics is erroneous, there are some inconsistencies and errors in the presented equations. For the applied model for the external calibration, it is not explicitly stated how this relates to the existing and published approaches. Comparisons are missing.

The numerical results presented in the Results section rely on a single reference dataset and a single part of the time series. Consequently, the dependence of the results from the chosen reference dataset remains open, such that it is quite hard to draw conclusions. The numerical test of the upward

continuation is not representative for the dataset used in the numerical analysis (in Sect. 3.2), as the real data set includes correlated data and systematic errors. Confirmation of the results with different datasets would be required to interpret the results correctly and reliable. Furthermore, it is not totally clear which version of the actual GOCE data were used.

The discussion and validation of the results is not clear. On the one hand, it is shown, that the trace-free criterion improves for the calibrated gradients. But, the other two chosen validation approaches are – from my point of view – not independent. If I understand the applied procedures correctly, the comparison using the TRF data is very similar to the first trace analysis and no additional data is involved. The higher order invariant analysis is – from my understanding – heavily biased, as consistency to the dataset used for calibration is checked. It is obvious and expected that the calibrated gradients are more consistent to the reference data compared to the un-calibrated gradients.

Finally, in the Conclusion chapter I am missing the actual conclusions. In the current form it is more a summary of the manuscript. But what follows from the performed study? Are the conclusions and results of the older studies still valid? How will the calibrated data be used? Are there jumps in the calibrated gradient time series?

To address this points more directly, please find below some more text specific comments to emphasize/justify my overall impression.

**2 Text specific comments**

**2.0 Title, Abstract, Keywords and General**

- *p. 1, "The ground gravity anomalies can...", l. 9*: The abstracts starts with a strange formulation. It sounds like that there is the one-and-only ground gravity anomaly.

- *p. 1, "...accuracy of the upward continuation method based...", l. 13*: This sounds as a very new technique is developed. But conversion of gridded data sets to spherical harmonics is a well established procedure. What is different to quadrature approaches as discussed for instance in Colombo (1981), Sneeuw (1994)? Why is it required to study the performance, it is quite well known, that this approaches work?

- *p. 1, "The traces of gravitational gradients after calibration were improved...", l. 20*: Please quantify the improvements in the abstract.

- *p. 1, "In addition, the relative errors of the tensor invariants...", l. 22*: Can you really conclude that? You compare to the same data set which you used for calibration. It must improve by definition!?!

- *p. 1, "...could meet the external calibration accuracy requirements...", l. 25*: What is the 'calibration accuracy requirement'? Where is this requirement defined?

- *p. 1, Abstract*: The abstract lacks a motivation for the study. Why is the external calibration required? For which purpose shall the external gradients be used? The context is missing.

**2.1 Introduction**

- *p. 1, l. 27ff, Introduction*: Within the introduction I am missing a motivation for this study. For which purpose you intend to use the externally calibrated gradients? What is your motivation?

- *p. 1, l. 36, "Therefore, the gravitational gradient..."*: Why is the band-limitation of the measurement signal responsible for the system errors? Please explain.

- *p. 1, l. 36ff, "The electrostatic gravity gradiometer..."*: I am missing a discussion of the general calibration strategy of the gravity gradients (e.g. Siemes et al. (2019), Siemes (2018)) applied in the L1B data set generation. How does this relate to your results? How does the improved L1B calibration changes the results/conclusions from previous external calibration studies?

- *p. 2, l. 45, "...in Bouman et al. and Pail (Pail, 2002;Bouman and Koop, 2003)."*: Please correct, in the text you refer to 'Bouman et al.', in the reference to 'Bouman and Koop'. Please be consistent.
- *p. 3, l. 79ff, "In this article, we discuss the possibilities..."*: The introduction misses an outline of the manuscript.

**2.2 Methods**

- *p. 3, l. 85, "A square integrable function..."*: I do not think that Kern, 2003 is a good reference for this statement.
- *p. 3, l. 85f, "A square integrable function..."*: A clear reference for the use gridded data to spherical harmonics conversion/representation is missing. Does the entire subsection refer to Colombo (1981)? Is there anything new? How does it relate to the quadrature approaches discussed in Sneeuw (1994)?
- *p. 3, l. 98, "...$\Delta_{ij} = \Delta\theta\Delta\lambda$..."*: Why does $\Delta_{ij}$ depend on indices $i$ and $j$?
- *p. 3, l. 98, "Combining Equations (3) and (4)..."*: How does this combination work? The result presented as eq. (5) is wrong. The results are the same for sine and cosine coefficients. The coefficients are independent of $m$ and the Legendre polynomials, longitude etc.
- *p. 3, l. 98, "Combining with the precise scientific..."*: How is the maximal degree of the spherical harmonic expansion chosen? How does the maximum degree relate to $N$?
- *p. 5, Fig. 5*: Where is the figure taken from? It is very similar to Fig. 8-1 in EGG-C (2010). Add the reference which was used to crate the figure/where the figure was taken/modified from. In addition, Fig. 1 contains much more details then required and quantities not used in the manuscript.
- *p. 4, Section 2.2*: Is this not just a simplified model compared to Bouman et al. (2004)? What is the difference? Please relate the used external calibration model to existing and published models/approaches.
- *p. 4, eq. (7)*: Above you used the symbol $\lambda$ for the longitude. Furthermore, $\lambda$ and $b$ require to have an index $_{ij}$ as well, or is it the same for all tensor components? Furthermore, in this methods section the segmentation discussed later is not discussed. Especially for the transition from one segment to the next this is rather important.
- *p. 4, l. 131, "$V_{ij}^s$ are the GOCE gradiometry observations"*: More precision is required here. I guess $V_{ij}^s$ are the bandpass filtered gradiometry observations? Filtering is not mentioned in the methods section. Furthermore, $V_{ij}^m$ denotes the filtered upward continuation values?
- *p. 4, l. 132, "Least squares estimation was then used to estimate the parameters."*: What about data screening? Outlier detection, etc?

**2.3 Results and Discussions**

- *p. 5, l. 137ff, Sect. 3.1*: What is the added value of the entire simulation performed in Sect. 3.1? It is well known, that the conversion of gridded data to spherical harmonics works quite well. Especially in your idealized scenario, where you synthesize ground gravity data from a model, the spherical harmonic coefficients shall be recovered within numerical floating point accuracy (your values actually seem quite high for scheme 1). I would expect that the more relevant parts for the external calibration are (a) the systematic errors in the ground gravity datasets, (b) the inhomogeneous data quality of such global datasets, (c) the correlations of the ground data due to the gridding processes applied in the dataset generation, etc. This aspects will significantly affect the conversion to spherical harmonics and thus the calibrations. None of this aspects are studied/captured by the performed numerical simulation, which covers just consistent signal content and white noise. Compared to the gain, this part covers three pages, which is quite long.

- *p. 5, l. 139, "The high-precision global gravity..."*: What classifies EGM2008 as high-precision gravity field model? Why is this important for the study?

- *p. 5, l. 139, "Pavlis et al., 2008"*: There is a much better citable reference for EGM08, it is Pavlis et al. (2012).

- *p. 5, l. 143, "Scheme 1. The grid gravity..."*: What accuracy would you expect? I would – in the noise free consistent scheme – expect that the recovery of the spherical harmonic coefficients works within numerical floating point precision.

- *p. 5, l. 149, "Scheme 2. The gravity anomalies..."*: As discussed above, I do not see why the white noise examples are required. Whereas scheme 1 can be seen as a validation of the software, the different white noise scenarios do not provide further information compared to what is expected from the real data conversion.

- *p. 5, l. 153, "There is no general pattern can be observed,..."*: Please reformulate.

- *p. 5, l. 157, "Therefore, the accuracy of the ground gravity anomalies...*: What is the consequence of this conclusion?

- *p. 5, l. 172, "Moreover, if the ground data are more accurate,...*: Is this really an important point? A global gravity anomaly data set with white noise more accurate than $1\,\mathrm{mGal}$ to $2\,\mathrm{mGal}$? If those data sets were required to calibrate GOCE observations, what would be the added value of GOCE?!?

- *p. 8, l. 180, "Compared with the global gravity field model,..."*: How can this be concluded? Where does this statement result from?

- *p. 8, l. 181, "Therefore, the DTU13 global..."*: Why DTU13 and not a global model e.g. a static global GRACE model? For the calibration, I would expect the long wavelengths to be of highest importance. Why do you focus on the high frequency signals? I am quite sure, that global models based on GRACE are much more accurate compared to DTU13 for the longer wavelengths. Have you compared your results calibrating with respect to a GRACE global model? What happens to your results when you change the input data set, are the results comparable? What would happen in case you calibrate against a global GOCE-only model? I have the feeling that the results strongly depend on the quality of the used data set. A comparison of calibration parameters derived from different datasets would be beneficial to resolve such doubts.

- *p. 8, l. 181, "...and its accuracy is about $2\,mGal$..."*: Where does this number come from? Have you verified?

- *p. 8, l. 184, "...the remove-restore procedure was applied..."*: Why is this suddenly required? In the methodological section you have not discussed this. In theory, there should be no difference. Have you verified/confirmed your results for any other part of the GOCE time series?

- *p. 8, l. 187, "The GOCE data used in this study..."*: Which version of the GOCE L1B/L2 data set has been used? There have been many reprocessing campaigns with respect to the calibration procedure. In the context of calibration, it is very important to be precise here.

- *p. 8, l. 188, "...February 11 to June 23, 2011..."*: What is the reason for choosing exactly this period?

- *p. 8, l. 188, "Referring to Bouman et al. (Bouman et al., 2011), the calibration period was set to 7 days."*: Since 2011 the underlying level 1B data changed significantly, can you confirm that 7 days is still a reasonable choice? How can you verify this? What about edge and border effects at the transition from one calibration period to the next? Is there an overlap to guarantee a smooth transition? If not, the independently estimated biases will introduce jumps in the calibrated gravity gradients. That might be an artificial systematic error source, which can be problematic, e.g. when you use this calibrated date for gravity field recovery.

- *p. 8, l. 193, "The outliers were replaced by cubic spline interpolation..."*: Why is a replacement of the outliers important? For the calibration you can simply discard the suspicious observations?!?

- *p. 8, l. 194, "The power spectral density (PSD)..."*: Did you use the most recent GOCE data release? From the characteristics in the PSD, it seems that it is one of the older releases which

was of lower quality below the measurement band. But the plot can of course be misleading as well.

- *p. 8, l. 194, "They pointed out that the cut-off frequencies..."*: Again, the conclusions of Bouman et al. (2011) are valid for the very old L1B gravity gradients, which were improved several times since then (e.g. Siemes et al., 2019, Stummer, 2013, Stummer et al., 2012, 2011). I am not sure if this conclusions are still valid for the newer gradient data sets of higher quality in the longer wavelengths.

- *p. 9, l. 214, "It appears that the scale factors..."*: Can you verify this with a different part of the time series? Can you verify this with a different reference dataset to calibrate against? It would be essential to demonstrate that the estimated scale factors do not strongly depend on the chosen reference dataset. Why not using spherical harmonics models directly? What happens if you use a GOCE-only global spherical harmonic model for the calibration purpose? Are all scale factors estimated as one in such a case?

- *p. 9, l. 216, "...the biases were very small..."*: Are the estimated biases statistically significant?

**2.4  Results and Discussions**

- *p. 11, l. 247, "It is obvious that the trace..."*: What do you think, is the trace reduction due to the scaling or due to the biases? What happens if you just estimate scale factors? What happens if you choose another reference dataset for calibration?

- *p. 12, l. 257, "The EGG_TRF_2 observations were transferred from the LNOF to the GRF..."*: What additional information to you expect to rotate the TRF gradients back to GRF? You already analyzed the trace of the measurements in the GRF. You basically undo the product generation of the TRF product and perform the same comparison as before? What do you expect from this compared to the previous test?

- *p. 12, l. 269, "Verification by the tensor invariants method"*: I do not get how this can be used as independent validation. It is obvious, that if you calibrate against the reference gradients, the consistency to invariants computed from the same reference gradients is larger. It should be the case by definition. So what can you conclude from this test?

**2.5  Conclusions**

- The provided conclusions are more or less a summary. But what can you conclude from the study? How to use the calibrated gradients? What are the next steps?

**References**

J. Bouman, R. Koop, C. C. Tscherning, and P. Visser. Calibration of GOCE SGG data using high–low SST, terrestrial gravity data and global gravity field models. *Journal of Geodesy*, 78(1-2): 124–137, September 2004. ISSN 0949-7714, 1432-1394. doi: 10.1007/s00190-004-0382-5. URL http://link.springer.com/article/10.1007/s00190-004-0382-5.

Johannes Bouman, Sophie Fiorot, Martin Fuchs, Thomas Gruber, Ernst Schrama, Christian Tscherning, Martin Veicherts, and Pieter Visser. GOCE gravitational gradients along the orbit. *Journal of Geodesy*, 85(11):791, November 2011. ISSN 0949-7714, 1432-1394. doi: 10.1007/s00190-011-0464-0. URL http://link.springer.com/article/10.1007/s00190-011-0464-0.

Oscar L. Colombo. Numerical Methods for Harmonic Analysis on the Sphere. Technical Report 310, Columbus, Ohio, 1981.

EGG-C. GOCE Standards 3.2. Technical report, 2010. URL http://www.earth.esa.int/GOCE/.

Nikolaos K. Pavlis, Simon A. Holmes, Steve C. Kenyon, and John K. Factor. The development and evaluation of the Earth Gravitational Model 2008 (EGM2008). *Journal of Geophysical Research: Solid Earth*, 117(B4):B04406, April 2012. ISSN 2156-2202. doi: 10.1029/2011JB008916. URL `http://onlinelibrary.wiley.com/doi/10.1029/2011JB008916/abstract`.

Christian Siemes. Improving GOCE cross-track gravity gradients. *Journal of Geodesy*, 92(1):33–45, January 2018. ISSN 0949-7714, 1432-1394. doi: 10.1007/s00190-017-1042-x. URL `https://link.springer.com/article/10.1007/s00190-017-1042-x`.

Christian Siemes, Moritz Rexer, Anja Schlicht, and Roger Haagmans. GOCE gradiometer data calibration. *Journal of Geodesy*, 93(9):1603–1630, September 2019. ISSN 1432-1394. doi: 10.1007/s00190-019-01271-9. URL `https://doi.org/10.1007/s00190-019-01271-9`.

Nico Sneeuw. Global spherical harmonic analysis by least-squares and numerical quadrature methods in historical perspective. *Geophysical Journal International*, 118(3):707–716, September 1994. ISSN 0956540X, 1365246X. doi: 10.1111/j.1365-246X.1994.tb03995.x. URL `https://academic.oup.com/gji/article-lookup/doi/10.1111/j.1365-246X.1994.tb03995.x`.

C. Stummer. *Gradiometer Data Processing and Analysis for the GOCE Mission*. PhD thesis, Institute for Astronomical and Physical Geodesy, Technische Universität München, Munich, Germany, 2013. URL `http://nbn-resolving.de/urn/resolver.pl?urn:nbn:de:bvb:91-diss-20121123-1111698-0-3`.

C. Stummer, T. Fecher, and R. Pail. Alternative method for angular rate determination within the GOCE gradiometer processing. *Journal of Geodesy*, 85(9):585, September 2011. ISSN 0949-7714, 1432-1394. doi: 10.1007/s00190-011-0461-3. URL `http://link.springer.com/article/10.1007/s00190-011-0461-3`.

C. Stummer, C. Siemes, R. Pail, B. Frommknecht, and R. Floberghagen. Upgrade of the GOCE Level 1b gradiometer processor. *Advances in Space Research*, 49(4):739–752, March 2012. ISSN 0273-1177. doi: 10.1016/j.asr.2011.11.027.

---

## Author Comment (AC1)

Dear reviewer:

Thank you very much for your comments on our paper. We have carefully revised the manuscript according to your suggestion. Our responses to the comments are listed below:

**Comment 1:** To make it clear, please use the same color bar in Fig.2 and the same bound for vertical axis (PSD) in Fig.4.

**Reply:** We have modified the Fig.2 and Fig.4.

[revised manuscript text omitted]